# Preparing Well for Esophageal Endoscopic Detection Using a Hybrid Model and Transfer Learning

**DOI:** 10.3390/cancers15153783

**Published:** 2023-07-26

**Authors:** Chu-Kuang Chou, Hong-Thai Nguyen, Yao-Kuang Wang, Tsung-Hsien Chen, I-Chen Wu, Chien-Wei Huang, Hsiang-Chen Wang

**Affiliations:** 1Division of Gastroenterology and Hepatology, Department of Internal Medicine, Ditmanson Medical Foundation Chia-Yi Christian Hospital, Chiayi 60002, Taiwan; vacinu@gmail.com; 2Obesity Center, Ditmanson Medical Foundation Chia-Yi Christian Hospital, Chiayi 60002, Taiwan; 3Department of Mechanical Engineering, National Chung Cheng University, Chiayi 62102, Taiwan; nguyenhongthai194@gmail.com; 4Division of Gastroenterology, Department of Internal Medicine, Kaohsiung Medical University Hospital, Kaohsiung Medical University, Kaohsiung City 80756, Taiwan; fedwang@gmail.com; 5Department of Medicine, Faculty of Medicine, College of Medicine, Kaohsiung Medical University, Kaohsiung City 80756, Taiwan; minicawu@gmail.com; 6Graduate Institute of Clinical Medicine, College of Medicine, Kaohsiung Medical University, Kaohsiung City 80756, Taiwan; 7Department of Internal Medicine, Ditmanson Medical Foundation Chia-Yi Christian Hospital, Chiayi 60002, Taiwan; cych13794@gmail.com; 8Department of Gastroenterology, Kaohsiung Armed Forces General Hospital, Kaohsiung City 80284, Taiwan; 9Department of Nursing, Tajen University, 20, Weixin Rd., Yanpu Township, Pingtung 90741, Taiwan; 10Hitspectra Intelligent Technology Co., Ltd., Kaohsiung City 80661, Taiwan

**Keywords:** endoscopy anatomy, artificial intelligence, deep learning, Vision Transformers, transfer learning

## Abstract

**Simple Summary:**

The timely detection and accurate classification of esophageal cancer are critical for providing optimal treatment. However, assessing and categorizing pathological conditions related to the esophagus face limitations as they rely on reference document photo-documentation, and the accuracy heavily relies on the endoscopist’s expertise. In recent times, computer-aided endoscopic image classification has achieved remarkable success in this domain. For this study, a dataset of 1002 endoscopic images, comprising 650 white-light images and 352 narrow-band images, was collected for training. The esophageal neoplasms were categorized into three groups: squamous cell carcinoma, high-grade dysplasia, and normal cases. To enhance the prediction results, a hybrid model was proposed, yielding an impressive accuracy of 96.32%, precision of 96.44%, recall of 95.70%, and f1-score of 96.04%. The introduction of AI-based diagnostic platforms is expected to effectively support medical professionals in formulating well-informed treatment regimens.

**Abstract:**

Early detection of esophageal cancer through endoscopic imaging is pivotal for effective treatment. However, the intricacies of endoscopic diagnosis, contingent on the physician’s expertise, pose challenges. Esophageal cancer features often manifest ambiguously, leading to potential confusions with other inflammatory esophageal conditions, thereby complicating diagnostic accuracy. In recent times, computer-aided diagnosis has emerged as a promising solution in medical imaging, particularly within the domain of endoscopy. Nonetheless, contemporary AI-based diagnostic models heavily rely on voluminous data sources, limiting their applicability, especially in scenarios with scarce datasets. To address this limitation, our study introduces novel data training strategies based on transfer learning, tailored to optimize performance with limited data. Additionally, we propose a hybrid model integrating EfficientNet and Vision Transformer networks to enhance prediction accuracy. Conducting rigorous evaluations on a carefully curated dataset comprising 1002 endoscopic images (comprising 650 white-light images and 352 narrow-band images), our model achieved exceptional outcomes. Our combined model achieved an accuracy of 96.32%, precision of 96.44%, recall of 95.70%, and f1-score of 96.04%, surpassing state-of-the-art models and individual components, substantiating its potential for precise medical image classification. The AI-based medical image prediction platform presents several advantageous characteristics, encompassing superior prediction accuracy, a compact model size, and adaptability to low-data scenarios. This research heralds a significant stride in the advancement of computer-aided endoscopic imaging for improved esophageal cancer diagnosis.

## 1. Introduction

Barrett’s esophagus is a condition affecting the esophagus, characterized by the appearance of lesions in the lower third of the esophagus [1]. The underlying cause of these injuries remains unclear, but some studies suggest that acid reflux from the stomach into the esophagus may lead to the deterioration of the esophageal wall’s lining tissue, resulting in Barrett’s esophagus [2,3,4,5]. The esophagus, a tubular smooth-muscle organ, is composed of three primary muscle layers: the mucosa, the muscular layer, and the connective tissues. Normally, the mucosa’s cell layer exhibits a pinkish-white color. However, in the case of Barrett’s esophagus, the cells in this lining undergo alterations, adopting a red hue resembling the mucosal cells of the stomach [6]. It is important to note that Barrett’s esophagus itself is not esophageal cancer. Nonetheless, the cells in Barrett’s esophagus possess a propensity to progress into dysplasia, a condition deemed precancerous due to its high potential for cancerous growth [7,8]. Dysplasia in Barrett’s esophagus varies in severity, ranging from mild to severe dysplasia. As dysplasia becomes more severe, the risk of malignant transformation into cancer increases [9]. The evaluation of esophageal dysplasia involves a histopathological examination of tissue samples. Low-grade dysplasia is identified when only a few cells display early signs of pre-cancerous growth. On the other hand, in cases where the esophageal cell sample exhibits significant abnormalities, it is categorized as high-grade dysplasia (HGD). HGD serves as a critical marker for predicting the potential development of precancerous esophageal cancer.

Esophageal cancer is a malignant disease in which tumors develop from the epithelial cells of the esophagus, disrupting the orderly division of cells and forming tumors [10,11]. It is one of the most prevalent types of gastrointestinal cancer. Esophageal cancer presents different disease manifestations at each stage, making early detection challenging, and diagnosis and treatment are typically performed when the disease has advanced. There are two primary types of cancer cells originating from the esophagus: squamous cell carcinoma (SCC) and adenocarcinoma [11]. SCC typically arises from the squamous cells that line the inner surface of the esophagus.

Esophagogastroduodenoscopy (EGD) is an examination approach in the diagnosis of diseases related to esophageal cancer, gastric cancer, or abnormal symptoms related to esophageal disease, esophageal precancerous lesions, and superficial cancer. Therefore, diagnosis based on endoscopic photo-documentation images is an important source of information, greatly influencing doctors’ analysis, the conclusion of pathology, and an appropriate treatment direction. Diagnosis from endoscopic images is usually based on the invasion depth of SCCs. Based on the micro-vessel morphology classification, the appearance and size of endoscopic lesions are abnormally small tumors that are cancerous or have other inflammatory symptoms [12,13]. However, clinical manifestations are often confused with other symptoms of esophagitis, such as acid reflux from the stomach, Barrett’s esophagus, or inflammatory manifestations of esophageal cancer.

The clinical diagnosis of Barrett’s esophagus and esophageal cancer relies on the assessment of EGD. However, distinguishing between the manifestations of Barrett’s esophagus and esophageal cancer on EGD images can be challenging, as both conditions involve lesions appearing in the outer lining of the esophagus. Notably, HGD in Barrett’s esophagus often exhibits pathological characteristics similar to SCC esophageal cancer. This similarity in pathological features underscores the complexity in accurately differentiating between the two conditions solely based on endoscopic imaging [14]. As a result, precise diagnosis and differentiation between Barrett’s esophagus and esophageal cancer necessitate a comprehensive histopathological evaluation and careful consideration of clinical findings. The quality assurance of EGD procedures primarily relies on reference document endoscopic photo-documentation, which demands a high level of expertise in endoscopic diagnostic imaging. As this process often involves the manual evaluation of each image based on image data, errors can accumulate over time. Therefore, it can be challenging to make diagnostic conclusions based solely on endoscopic photo-documentation to differentiate between HGD and SCC cancer levels.

White-light imaging (WLI) is a valuable tool in EGD photo-documentation for investigating esophageal pathology. However, up to 40% of esophageal precancerous lesions are misdiagnosed using WLI [15,16]. To overcome these problems, narrow-band imaging (NBI) is recommended to improve the WBI’s flawed prediction rate for esophageal precancerous lesions and superficial cancer [17]. NBI utilizes digital optics that filter light into narrow spectra, exploiting the light’s penetration properties. Short-wavelength rays easily penetrate the superficial mucosa, while long-wavelength rays penetrate deeper into the mucosa [18,19]. This technique enhances endoscopic images and improves the contrast between mucosa and superficial micro-vessels, making them appear as dark brown structures against a pale green background of internal organs [13]. Figure 1 depicts examples of WLI and NBI with different esophageal lesions, including normal, HGD, and esophageal cancer, i.e., SCC. It is evident that the pathological features of the epithelium are not apparent in the endoscopic images in cases of HGD (Figure 1b,e) and SCC (Figure 1c,f), leading some endoscopists to experience difficulty discriminating them.

With the emergence of artificial intelligence (AI), computer-aided diagnosis is gradually becoming an indispensable aid for endoscopic image diagnosis, especially for EGD. A number of studies have shown the application of AI in supporting pathological diagnosis from endoscopic images. Many studies have used deep learning methods in addressing esophageal lesion problems. Numerous approaches have been proposed to address endoscopic image processing problems, including classification, segmentation, or object detection methods. Yamada et al. [20] developed a real-time endoscopic image diagnosis system for detecting the early symptoms of colorectal cancer through colonoscopy. Chang et al. [21] proposed a method to identify EGD according to the classification of endoscopic anatomy images; the author used the ResNeSt network [22], which is a combination of the CNN backbone network Resnet50 [23] and Vision Transformer structure [24]. Instead of using a ResNeSt network, we combined EfficientNet and Vision Transformer networks. Wang et al. [25] proposed a deep learning model for the image classification of endoscopic gastroesophageal reflux disease. However, the transfer learning approach based on the model VGG-16 [26] makes the accuracy of the prediction results on the test set only 87.9%. These prediction results are not too outstanding, because these approaches all use built-in object detection models with specialized purposes to identify objects that often appear in nature and life, which is inadequately effective when applied to medical images. Yu et al. [27] provided a multitask model that uses a combination of classification and segmentation models with an attention mechanism. Wu et al. [28], Pan et al. [29], and Celik et al. [30] also proposed multitask models to identify esophageal lesions. However, these approaches are unnecessary when targets need to be differentiated and indicating the exact location of a tissue is not needed.

In this study, we develop an expert system to serve the medical imaging community. We proposed a solution to detect three types of diseases related to esophageal disease, namely, HGD, SCC, and normal cases, by using an integrated deep learning technique, which combines many of today’s advanced and high-performance models, namely, EfficientNet [31] and Vision Transformer [24], to address the limitations of WLI images and exploit the advantages of NBI and obtain the highest prediction accuracy. Transfer learning techniques address low-accuracy predictions due to the limited amount of data and large number of patients and policy regulations for using medical images in research. We configure each hybrid model using different transfer learning methods, namely ImageNet, AdvProp, and Noisy Student. We assess the performance of these transfer learning strategies when applied to a set of custom data. Our contributions are highly accurate compared with those of architectural models that constitute it, and the current standout models are Resnet [23] and ResNeSt [22].

In this study, our contributions are as follows:Hybrid model results in high accuracy while ensuring compactness when making real-time predictions;Transfer learning can handle data shortage, reduce data collection time, and deploy the application for many different custom dataset cases;Visualization is provided to a model, especially a model based on the attention mechanism, and support for visual diagnosis using images is provided.

## 2. Materials and Methods

### 2.1. Data Preparation

In the development of a system for the diagnosis of esophageal neoplasms, 46 patients with esophageal disease are recruited from Kaohsiung Medical University Hospital, and imaging data are collected. The patient cohort comprises 10 patients with Barret’s esophagus without dysplasia, 20 patients diagnosed with HGD of Barret’s esophagus, and 16 patients with esophageal SCC. The diagnosis is mainly based on the evaluation of specialist pathologists. A total of 1002 images are collected for training, which are classified into three categories: normal esophagus (Normal; 202), high-grade dysplasia (HGD; 498), and SCC (302). A total of 650 and 352 images for WLI and NBI are obtained, respectively. To evaluate the effect of NBI on medical image prediction, the data are divided into three main groups for training: WLI only, NBI only, and a combination of WLI and NBI images. The prediction results are then used to draw conclusions and prepare the data for this esophageal endoscopic imaging problem. The objective is to enrich the diversity of the dataset, improve detector performance with the additional knowledge, and minimize the number of false positive detection results.

The images are resized to 224 × 224 × 3 for both the training and testing phases. Pre-processing steps are applied to fit the Pytorch platform, including normalization with a mean value of (0.485, 0.456, 0.406) and a standard deviation value of (0.229, 0.224, 0.225). In order to enrich the datasets, we employ algorithms for data augmentation, with the selection of methods based on the references [32,33,34]. The most popular and effective data augmentation operations for CNNs are crop and flip for Resnet, and cutout, crop, and flip for ResNeSt. A platform pipeline is shown in Appendix A. It has two main parts, namely, the training phase and deployment phase. In the training phase, data are managed and automatically updated and downloaded during training via a cloud database. Data are automatically downloaded during training, and the best-trained weights are automatically updated in the system. In the deployment phase, a user interface is developed to display the parameters on the screen.

### 2.2. Network Architecture

Utilizing empirical experience, we select the EfficientNet-B0 model for the backbone as a feature extraction structure. As shown in Appendix A, EffiecientNet-B0, although not the most accurate model in the EfficientNet family, is the simplest variant model and can fit training device resources and display execution devices, such as monitors of endoscope systems. The output of the last layer of encoder module EfficientNet-B0 is 7 × 7 × 320.

The Vision Transformer (ViT) [24] is a construct used in computer vision, based on the Transformer [35] architecture originally designed for natural language processing tasks. Inheriting the principles of the Transformers structure, we split input feature vectors into N patches (in our architecture, N = 7 × 7) and flatten them to 49 embedding vectors. An input feature vector of size N × N (7 × 7) is split into N/M^2^ (7/1) patches of size M × M. Each patch is called a “token”. Dividing into patches makes object recognition challenging for the model but increases the spatial awareness of objects’ features. Position embeddings are applied to patches. These learnable position-embedding vectors including N × N patches and a class vector are added to patch-embedding vectors as Transformer inputs. As shown in Appendix A, the structure of the hybrid model constructs the backbone EfficientNet as an encoder module structure and the Vision Transformer as a decoder module. The encoder module is responsible as a feature extractor, performing the inheritance of features learned from large databases and transferring learning as the input for the decoder module. The Decoder module takes advantage of these representative mechanisms of the Transformer such as position encoded, the multi-layer perception head structure, and the attention mechanism to optimize the learning of inherited features. The learned weights are then passed to the Classifier module to perform the classification of esophageal pathology: dysplasia, SCC, and normal, respectively.

The classifier module is fine-tuned to ensure adherence to the classification requirements. In this particular classifier module, a new fully connected (FC) layer head is constructed and then the FC layer head and the body of the network are trained concurrently. The batch normalization layers with mean and standard deviation values obtained from the trained weight on the ImageNet dataset will be frozen during the training phase. The classifier module is structured according to the series of layers, and each layer has a new structure: FC followed by ReLU and Dropout layers. Specifically, Architectural parameters of the classifier module are as follows: Linear layer (embedded_size = 320, 256), followed by ReLU and Dropout Layer (0.1), Linear layer (256, 128) followed by ReLU and Dropout (0.1), and Linear layer (128, num_classes = 3).

### 2.3. Experimental Setups

The Adam optimizer is used with an initial learning rate of 1 × 10^−4^ and weight decay of 0.00001, and the batch size is set at 6. Cross-entropy loss is used for the optimization of the network. All experiments are conducted on a Windows 10 machine with the following specifications: Intel i5-7400 @ 3.00 GHz, GPU NVIDIA GeForce GTX 1070, 32 GB RAM. The network is implemented with Pytorch framework. The network is trained for 100 epochs. In taking advantage of pretrained weights from a large source task, changing weights excessively and rapidly will affect the efficiency of training. In the classification task, the cross-entropy loss function is given by the following formula:$$L_{C}=-\frac{1}{n}\sum_{i=1}^{n}\left[y_{i}\log\left({\hat{y}}_{i}\right)+\left(1-y_{i}\right)\log\left(1-{\hat{y}}_{i}\right)\right],$$
where *n* denotes the number of examples, $y_{i}$ denotes the annotated examples, and ${\hat{y}}_{i}$ denotes the predicted examples.

### 2.4. Transfer Learning Strategies

Leveraging the pretrained weights from the aforementioned models can reduce the amount of training required and overcome the shortage of custom data. In computer vision, two types of transfer learning are commonly used: (1) using pre-trained networks as feature extractors, and (2) fine-tuning some fully connected layers of a CNN network. According to empirical experience, we choose to fine-tune the CNN network to perform transfer learning.

Transfer learning methods leverage pretrained weights from the massive ImageNet dataset [36], which contains over 1.4 million images across 1000 object classes, making it the most widely used database for existing models. Another notable semi-supervised learning technique is Noisy Student Training [37], which incorporates concepts from self-training and distillation by using equivalent or larger “student” models and injecting noise during training. The Noisy Student training strategy involves treating a student as a teacher to relabel unlabeled data and train the student. This approach offers several significant advantages, such as allowing students to grow larger than teachers, enabling them to learn better from a larger dataset. Additionally, adding noise to students forces them to learn from the pseudo-labels, further enhancing their performance.

Adversarial Propagation (AdvProp) [38] is a learning strategy that trains models using both clean and adversarial images simultaneously. This approach employs an auxiliary batch norm for adversarial images to prevent overfitting.

## 3. Results

### 3.1. Evaluating Transfer Learning Strategies

We evaluate the effectiveness of each transfer learning strategy by measuring the accuracy of the models. The hybrid models are configured with different EfficientNet backbones, designated from B0 to B7. The results of the prediction accuracy of hybrid models from B0 to B7 are shown in Figure 2. Particularly, in the case of WLI images as shown in Figure 2a, model B7 achieves the best performance with the AdvProp transfer learning strategy (97.58%). This suggests that AdvProp is better suited for this task, as the accuracy of models configured with backbones B3 to B7 consistently outperforms those of other transfer learning strategies. However, for the NBI image set as shown in Figure 2b, which is characterized by unbalanced data, the evaluation and selection of appropriate transfer learning strategies are less clear-cut. While the Noisy Student method yields accurate predictions for configured models B0, B1, B2, and B3, the AdvProp method shows excellent performance.

The WLI + NBI dataset, with a larger and more diverse set of learned features, shows superior prediction accuracy using the configured model B3 for the ImageNet transfer learning strategy, with an accuracy of 96.32% (Figure 2c). The ImageNet strategy also performs well in the configured models B2, B5, B6, and B7. The AdvProp strategy is effective in models B0, B1, and B4.

### 3.2. Model Performance Evaluation

After evaluating the suitability of transfer learning strategies based on their classification performance, we choose to implement the interface platform using the configuration model B3 with the AdvProp transfer learning strategy. The training and validation accuracy and losses are shown in Appendix A, and our proposed models achieve convergence very early from the 25th epoch, with the classification accuracy continuing to improve throughout training. The effectiveness of each class dataset of esophageal disease is evaluated based on accuracy, confusion matrix, and receiver operating characteristic (ROC) diagrams, as shown in Appendix A. The evaluation of models in terms of accuracy and confusion matrix relies on the key indicators of Precision and Recall. The trade-off between Precision and Recall often leads to models with either high Precision and low Recall or low Precision and high Recall. As a result, determining the superiority of a model becomes challenging, as it is unclear whether Precision or Recall should be given higher priority during evaluation. To address this challenge, we introduce a novel metric that combines both Precision and Recall, known as the f1-score. The f1-score is calculated as the harmonic mean of Precision and Recall, providing a more balanced assessment of both measures. In situations where Precision and Recall exhibit significant differences, the f1-score aids in making a more objective evaluation, enabling us to effectively assess the model’s performance in a comprehensive manner. By considering both Precision and Recall simultaneously, the f1-score enhances the interpretability and fairness of the model evaluation process, promoting a more informed decision-making process in model selection and optimization.

The model performances for WBI, NBI, and WBI + NBI are listed in Table 1. In the WLI dataset, the overall accuracy achieved is 96.77%, accompanied by macro average precision and weighted average precision values of 96.70% and 96.77%, respectively. Similarly, macro average recall and weighted average recall are 95.50% and 96.77%, while the corresponding macro average f1-score and weighted average f1-score stand at 96.05% and 96.73%, respectively.

In the dataset comprising NBI images exclusively, the total accuracy attained is 92.42%. Moreover, the macro average precision and weighted average precision are measured at 86.05% and 92.23%, respectively. The macro average recall and weighted average recall values are calculated as 83.00% and 92.42%, respectively. Additionally, the macro average f1-score and weighted average f1-score are determined to be 84.21% and 92.25%, respectively.

For the enhanced dataset that integrates both WLI and NBI images, the total accuracy obtained is 96.32%. In this case, the macro average precision and weighted average precision are found to be 96.44% and 96.32%, respectively. The macro average recall and weighted average recall are computed as 95.70% and 96.32%, respectively. Furthermore, the macro average f1-score and weighted average f1-score are revealed as 96.04% and 96.29%, respectively. A comparative analysis indicates that the performance of the enhanced dataset surpasses that of the previous dataset, mainly attributed to the utilization of a larger number of training images.

Figure 3 presents the confusion matrices for all datasets. In particular, for the WLI dataset, the AdvProp strategy is found to be suitable for the detection of the Dysplasia category, with 55 out of 57 correctly classified images. For the SCC class, the AdvProp strategy is more appropriate, as up to 41 out of 53 images are correctly classified. Although the difference in accuracy is not significant and the training time is not much different, we conclude that the AdvProp strategy is the best fit for the WLI dataset. Regarding the NBI image set, which is a group of unbalanced data, the evaluation and selection of appropriate transfer learning strategies are unsatisfactory. The AdvProp method demonstrates correct prediction results for SCC and Dysplasia classes. For the WLI + NBI dataset, the model shows excellent performance in classifying Dysplasia and SCC, with 100 out of 105 and 52 out of 59 images correctly classified, respectively.

Figure 4 presents the ROC curves for all datasets and three classes (0—Normal, 1—Dysplasia, and 2—SCC). For the WLI dataset, the AUC values range from 0.98 to 1.00, indicating high classification accuracy. Similarly, for the WLI + NBI dataset, the AUC values for all classes are 0.99, demonstrating excellent classification performance. However, for the NBI dataset, the AUC value for the “normal” class is slightly lower (0.78) due to the limited number of images in this class. The differences in classification results for the three classes are not significant, indicating that all transfer learning methods are effective in classifying esophageal endoscopic images on WLI and NBI.

### 3.3. Visualization of Hybrid Model

When interpreting the working mechanism of a CNN architecture, one of the simplest ways is to explore the intermediate activation layers. In the field of computer vision, these channel activations are explicitly interpretable and represented by 2D images. Such patterns can aid researchers in determining what their models have learned and suggest medical image features that facilitate diagnosis.

Unlike the linear structure of conventional CNN models, activation maps cannot be easily accessed in Vision Transformers, because they are governed by queries, values, and key weights. Vision Transformer is composed of encoding blocks with several attention heads that connect information from patches in images. The MLP block transforms the represented patches into higher-order feature representations. The attention heads and MLP blocks are connected by residual connections. In contrast to the Vision Transformer network that will receive an input weight × height × channel image, i.e., 224 × 224 × 3, the ViT block of our hybrid network will receive the feature output from the backbone block, that is, the pretrained EfficientNet-B0 network is 7 × 7 × 320 in size. Our hybrid network has eight attention heads. Therefore, the size of each attention head is 8 × 50 × 320, and each attention head will take over 50 tokens. Each token has a feature representation with a depth of 320. Among these 50 tokens, 49 tokens represent 7 × 7 feature patches, and one token represents a feature’s token class. Inside each attention head, three main weights represent a token: Query (Q), Key (K), and Value (V).

In the Transformer model, the self-attention mechanism is the aggregation of information from the input-embeddings layer to a subsequent layer. The flow of information is the amount of information propagated from the input layer to a higher layer where the update in embedding occurs. However, the flow of information affected by different tokens is gradually mixed, and thus attention weights seem difficult to capture and query. As shown in Appendix A, “the attention” of our model is focused on multiple sites; attention-deficit zones appear, such as the SCC zone; and some focus on inflammatory micro-vessel system variability. Some diagnostic areas appear repeatedly, without consistency, leading to the inability to draw specific conclusions about the location of defects in the esophageal canal. That causes inconsistencies in the model interpretation of pathological etiology, particularly the location of the tumor and the inflammatory mass. However, Appendix A only explains how individual activation layers are represented and does not explain how VIT focuses and how they arrive at predictive conclusions.

As shown in Appendix A, a method called “Attention Rollout” is proposed to quantify the activation layers into the most general layer by evaluating the weights, discarding less influential weights, and retaining an interest in the information flow in the higher layers [39]. Attention Rollout suggests taking the average of the heads and discarding a certain percentage of weights by removing noise according to the weights’ common denominator as the *Mean* fusion. However, we can take the minimum or maximum weights (*Min* or *Max* fusion, respectively). In this study, we observe that taking the maximum value produces the most intuitive results compared with average fusion and minimum fusion. Figure 5 shows the comparison of methods for calculating values among the attention heads, *Mean*, *Min*, and *Max* fusion. It can be seen that the feature maps show the most convergence and concentration at *Max* fusion, while the remaining algorithms, i.e., *Min* and *Mean* fusion, show the distribution of feature maps that are still fragmented and lacking in connection. Figure 6 shows the false predictions. Figure 6a,b show that the false category is not predicted, with a low probability of accuracy, 64.28% and 70.90%, respectively; and the result of feature maps also shows difficulty when the model is unable to indicate the location of tumor occurrence or abnormal deformities, instead of an arbitrary position in the image. In Figure 6c, although the label prediction is correct, the feature map results also have challenges. The cause is the duplicate in the features of the “normal” category with other pathological categories. Consequently, feature maps present the embarrassment of not being able to predict the tumor’s location, leading to absurd focus points represented on heatmaps.

### 3.4. Ablation Study

In this section, we study the importance of noise and summarize the ablations for the other components of our method. Our proposed method is mainly based on the EfficientNet family, and the pretrained model EfficientNet plays a significant role in the transfer learning strategy. To improve the learning ability of esophageal endoscopic image features, we construct a hybrid model by adding the Vision Transformer part after the backbone model. As shown in Table 2, our hybrid models increase precision, recall, and f1-score compared with original Vision Transformer architecture, specifically ViT_small_patch16_224, ViT_base_patch16_224, and ViT_large_patch16_224. Our hybrid models exhibit a great improvement compared to the backbone models EfficientNet-B2 and EfficientNet-B3. Our hybrid model with backbone EfficientNet-B2 increases the Precision from 0.9440 to 0.9541, the Re-call from 0.9473 to 0.9526, and f1-score from 0.9455 to 0.9520. When compared with EfficientNet-B3, the value of the f1-score further increases from 0.9563 to 0.9629, precision increases from 0.9589 to 0.9644, and recall increases from 0.9559 to 0.9570. Finally, our proposed model achieves the highest f1-score. This shows that the attention module in the Vision Transformer may help learn more features from the backbone model. However, the improvements in Precision, Recall, and f1-score compared to EfficientNet models are not significant. Therefore, it can be concluded that the Vision Transformer structure does not bring much improvement in learning the features of the endoscope image compared with the traditional CNN models. This indicates that the ViT can focus on cancer tissue feature information, which is useful for our detection tasks.

### 3.5. Comparison with Other Existing Studies

In this section, we review the most notable studies by comparing the performance between existing models and our hybrid models. The configurations of the models used for evaluation in this experiment are shown in Table 3, which reviews several methods. In comparison to other existing studies, the Resnet Family [23], i.e., Resnet18, Resnet34, and Resnet50, and the split-attention network ResNeSt [22], our model achieves a comparable performance.

To evaluate the efficiency in deployment, we measure the time prediction for each image and take the average. In Table 3, the performance evaluation of the model is evaluated by accuracy, model size, our model, and the prediction time per image. The goal is to investigate the efficiency of actual deployment compared to existing models. We determine the recognition speed through our ability to predict the number of images per second. Figure 7 shows the performance of the models, from H0 to H7 denoted for our hybrid models. The plot visualizes the speed and performance tradeoff. In other words, a better performance in accuracy prediction may lead to inference taking longer. The above results show that, although our hybrid models operate longer during deployment, they will prioritize diagnostic accuracy in medical systems.

## 4. Conclusions

In this study, we propose a method based on deep learning for the detection of SCC from esophageal endoscopic WLI and NBI images. By taking advantage of transfer learning in overcoming the shortage of custom datasets and applying the superior algorithms in computer vision to medical imaging, we formulate a solution platform that is effective and facilitates endoscopic diagnostic imaging. Our deep learning model takes advantage of accuracy, compactness, and inheritance through features learned from extensive data sources in the EfficientNet network. Moreover, the attention mechanism of the Vision Transformer structure is rotated smoothly and creates certain effects, removing barriers between the technology transfer from the natural language processing array and the computer vision segment. One of the advantages that our hybrid brings is creating an AI-based platform where accuracy is improved while the model remains compact and the computational resources and required input data are conserved. Relatively moderate collection can be easily deployed and replicated in medical image processing applications.

In our future work, we aim to develop a model for different medical imaging objects, such as vascular images, X-ray images, CT, MRI, and PET, and use it in other medical applications, such as atrium segmentation, cardiac detection, pneumonia classification, lung tumor segmentation, and liver tumor segmentation. Additionally, we aim to develop a model for mobile devices to increase users’ browsing experience.

## Figures and Tables

**Figure 1 cancers-15-03783-f001:**
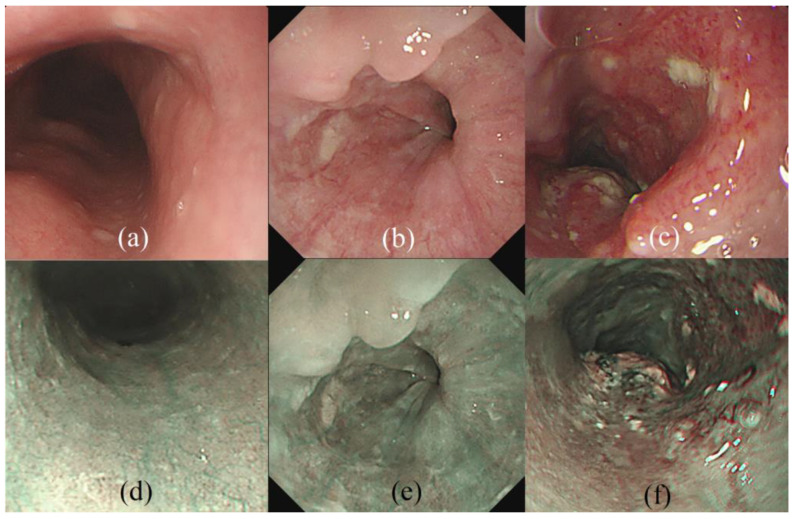
WLI examples with (**a**) normal, (**b**) HGD, and (**c**) SCC, and NBI examples of (**d**) normal, (**e**) HGD, and (**f**) SCC. The similarity of histopathological characteristics in endoscopic images between HGD and SCC makes it difficult for endoscopists.

**Figure 2 cancers-15-03783-f002:**
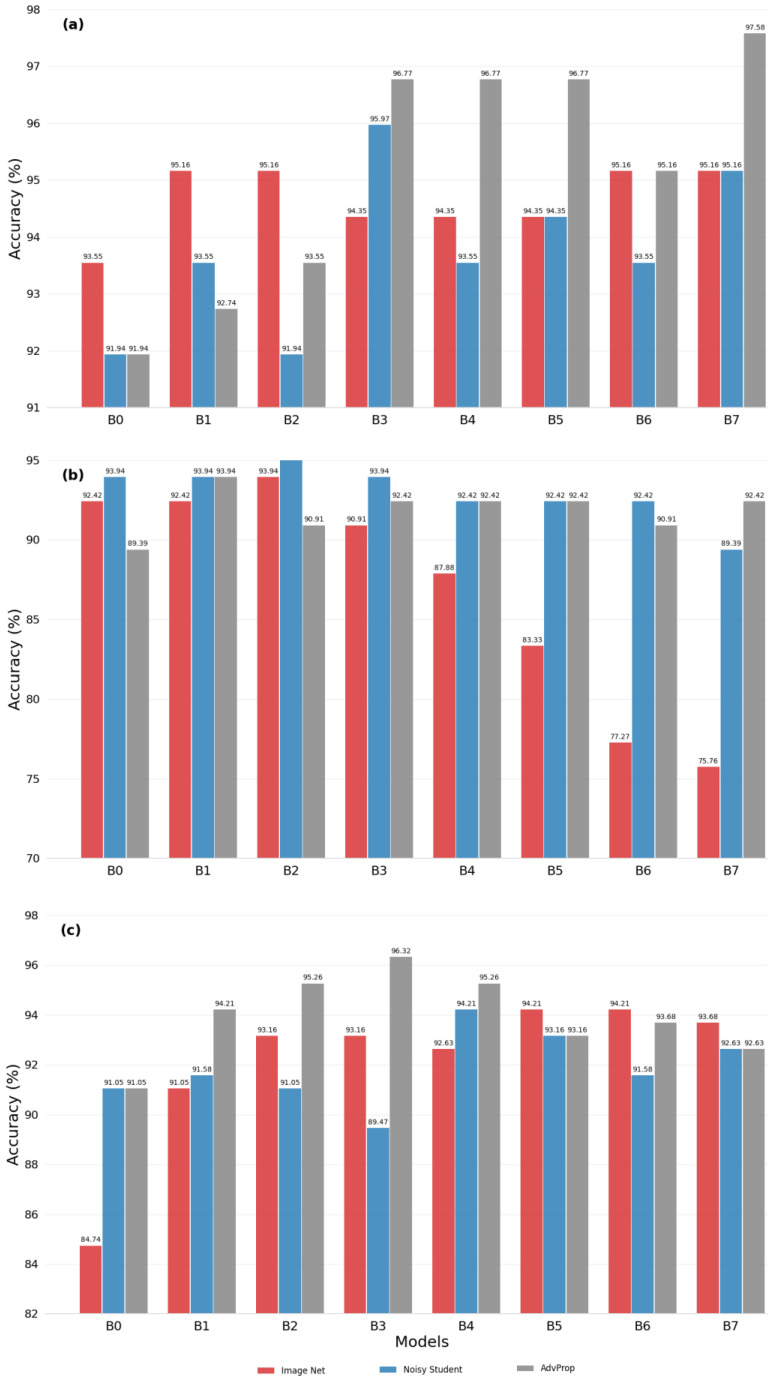
Accuracy for hybrid model with encoder module EfficientNet B0 ~ B7 with different datasets: (**a**) WLI, (**b**) NBI, and (**c**) WLI + NBI.

**Figure 3 cancers-15-03783-f003:**
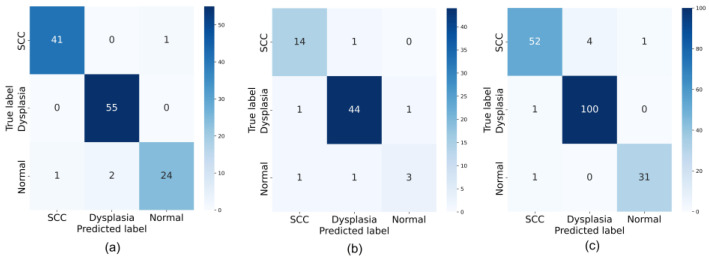
Confusion matrix for (**a**) WLI, (**b**) NBI, and (**c**) WBI + NBI dataset.

**Figure 4 cancers-15-03783-f004:**
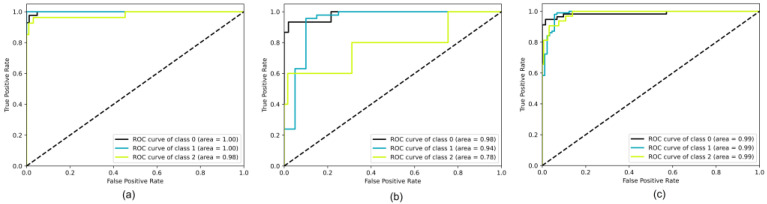
Receiver operating characteristics for (**a**) WLI, (**b**) NBI, and (**c**) WBI + NBI datasets.

**Figure 5 cancers-15-03783-f005:**
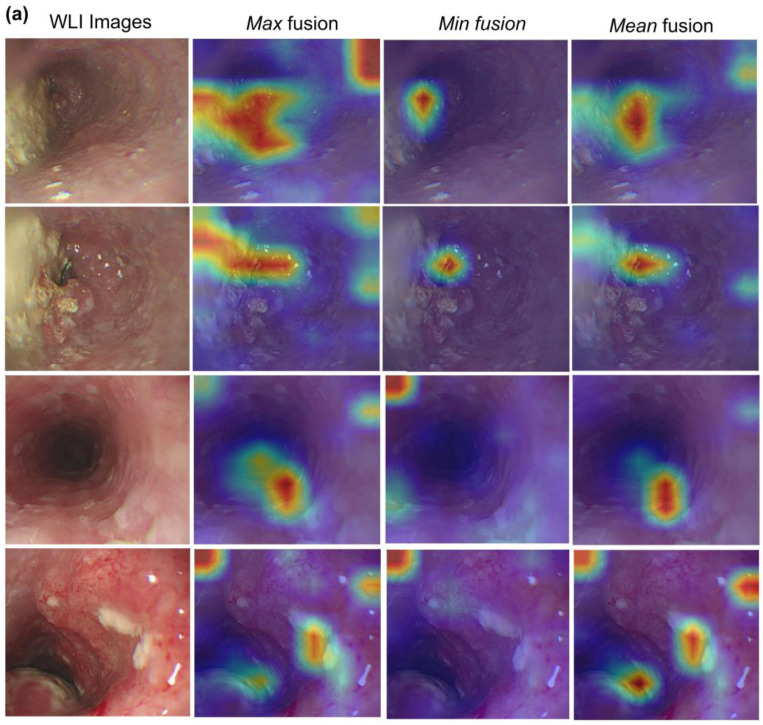
Visualization of Attention Rollout with different fusion heads with (**a**) WLI and (**b**) NBI images. Heatmap generated by Attention Rollout for model interpretation.

**Figure 6 cancers-15-03783-f006:**
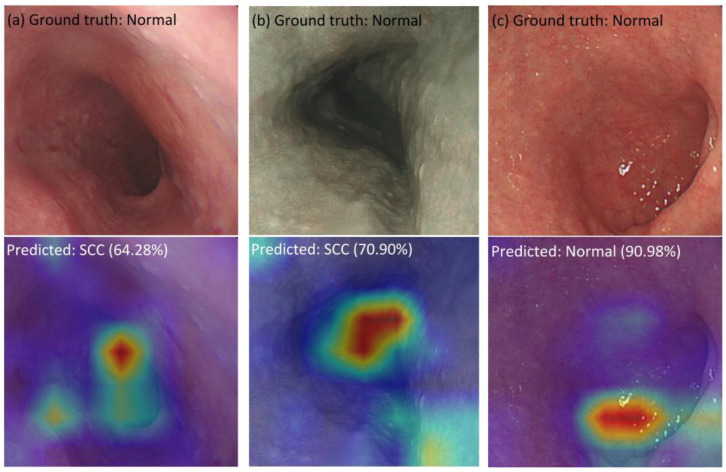
Examples of false predictions. Heatmap generated by Attention Rollout for model interpretation.

**Figure 7 cancers-15-03783-f007:**
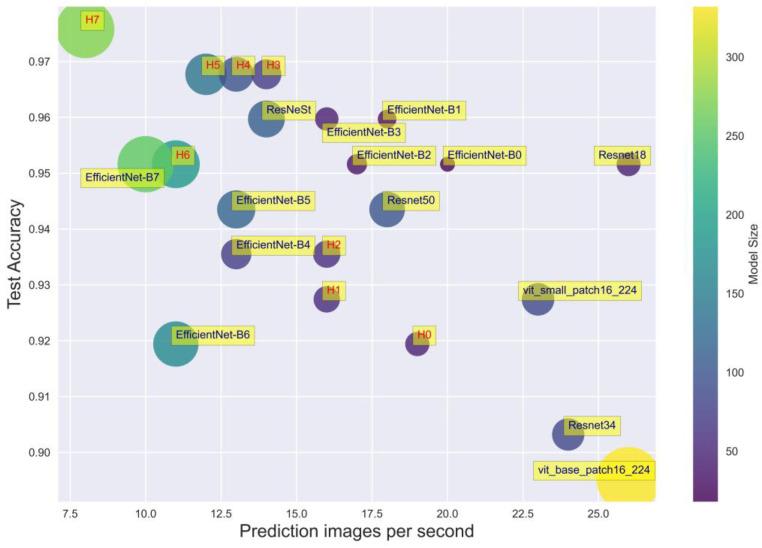
Comparison of model parameters and performances.

**Table 1 cancers-15-03783-t001:** Performance comparison among datasets containing WLI (without NBI images), containing NBI images only, and containing both WLI + NBI images.

	Metrics	All	Macro-Average	Weighted Average	Class		
					Normal	Dysplasia	SCC
WLI	Accuracy	96.77					
	Precision		96.70	96.77	97.62	96.49	96.00
	Recall		95.50	96.77	97.62	100.00	88.89
	f1-score		96.05	96.73	97.62	98.21	92.31
NBI	Accuracy	92.42					
	Precision		86.05	92.23	87.50	95.65	75.00
	Recall		83.00	92.42	93.33	95.65	60.00
	f1-score		84.21	92.25	90.32	95.65	66.67
WLI + NBI	Accuracy	96.32					
	Precision		96.44	96.32	96.30	96.15	96.88
	Recall		95.70	96.32	91.23	99.01	96.88
	f1-score		96.04	96.29	93.69	97.56	96.88

**Table 2 cancers-15-03783-t002:** Ablation experiments on the test dataset.

Model	Precision	Recall	f1-Score
ViT_small_patch16_224	0.9306	0.9084	0.9177
ViT_base_patch16_224	0.9077	0.8521	0.8652
ViT_large_patch16_224	0.8862	0.7849	0.8028
EfficientNet-B0	0.9535	0.9492	0.9512
Hybrid model (backbone EffiecientNet-B0)	0.9319	0.9352	0.9334
EfficientNet-B1	0.9511	0.9578	0.9539
Hybrid model (backbone EffiecientNet-B1)	0.9473	0.9473	0.9473
EfficientNet-B2	0.9440	0.9473	0.9455
Hybrid model (backbone EffiecientNet-B2)	**0.9541**	**0.9526**	**0.9520**
EfficientNet-B3	0.9589	0.9559	0.9563
Hybrid model (backbone EffiecientNet-B3)	**0.9644**	**0.9570**	**0.9629**

**Table 3 cancers-15-03783-t003:** Comparison of model parameters and performances.

Model	Number of Params	Size (MB)	Accuracy	Time Prediction per Image (s)
Resnet-18	12,084,267	46	0.7500	0.0380
Resnet-34	22,192,427	84	0.8710	0.0404
Resnet-50	26,738,219	102	0.9435	0.0553
ReStNet	28,664,427	109	0.9597	0.0697
EfficientNet-B0	4,795,519	18	0.9516	0.0481
EfficientNet-B1	7,301,155	28	0.9597	0.0538
EfficientNet-B2	8,554,501	33	0.9516	0.0576
EfficientNet-B3	11,615,275	44	0.9597	0.0617
EfficientNet-B4	19,341,616	74	0.9597	0.0745
EfficientNet-B5	30,389,784	117	0.9677	0.0741
EfficientNet-B6	43,040,704	165	0.9597	0.0836
EfficientNet-B7	66,347,960	254	0.9435	0.0925
ViT_small_patch16_224	22,379,883	85	0.9274	0.0434
ViT_base_patch16_224	87,093,483	332	0.8952	0.0376
ViT_large_patch16_224	304,983,531	1163	0.8468	0.0459
Hybrid model (backbone EffiecientNet-B0)	12,234,239	47	0.9194	0.0514
Hybrid model (backbone EffiecientNet-B1)	14,739,875	56	0.9274	0.0623
Hybrid model (backbone EffiecientNet-B2)	15,851,653	60	0.9355	0.0624
Hybrid model (backbone EffiecientNet-B3)	18,762,667	72	0.9677	0.0670
Hybrid model (backbone EffiecientNet-B4)	25,422,027	97	0.9677	0.0767
Hybrid model (backbone EffiecientNet-B5)	35,988,403	138	0.9677	0.0810
Hybrid model (backbone EffiecientNet-B6)	48,124,763	184	0.9516	0.0847
Hybrid model (backbone EffiecientNet-B7)	70,884,691	272	0.9758	0.1182

## Data Availability

The data presented in this study are available in this article.

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
