# Peer review of "Preparing Well for Esophageal Endoscopic Detection Using a Hybrid Model and Transfer Learning"

_cancers, 2023, doi:10.3390/cancers15153783_

Round 1
Reviewer 1 Report
Dear Sirs,
Thanks you so much for providing the opportunity to review the paper on "Well-preparation for esophageal endoscopic detection using a hybrid model and transfer learning" by Chou et al. covering endoscopic detection using and AI approach.
The topic is currently of great interest to the medical community and meets an important need.
However, I have some difficulties reading this paper and would suggest major revisions based on the following points:
Summary: Needs to be restructured. Not understandable
Abstract: break between line 43 and following: Hard to follow the thoughts of the authors. Difficult to read. What do the authors mean by reduce waste health system? This point is not further discussed later on.
Introduction: - Jumping a bit from Baretts' to different origins of esophageal cancer.
Why mention Baretts when its only focussing on SCC? Perhaps mention why only focus on SCC.
Line 142: actually just two types of dieseas and a control or three types of different tissues.
What is a f1-score? Define it in the text. And is there a difference between F1 and f1?
Is anonymisation a criteria for waiving informed consent?
Line 264-268: perhaps put this in the introduction
Line 274: what do you mean by personalization needs?
Please mention fig 2 in the text.
Please mention table 1 in the text.
Results paragraph 2 is hard to read
Paragraph 1 does not really state any results
Ablation Study 3.4: are the improvements significant?
Suggestions: Include some of the supplementary tables/figures as they may help to understand the paper better.
Line 71: quote
Line 111: what do you mean by cancer esophageal?
Line 112: It can 110 be seen that the pathological features of the epithelium appear are no
Line 180: normalize without a capital.
Lines 336/399: Normal without a capital or put ““
Line 249: result in
Line 348 ““?
Good luck with the revisions. Looking forward to reading the revised manuscript.
I would highly recommend an English nativ to read the paper.
Reviewer 2 Report
This article is interesting. The idea of artificial intelligence is well-known in colonoscopy, either screening or therapeutic. Much work is needed to use AI in GI endoscopy better and limit the false positive results. The word fusion needs more clarification. Compared to biopsy, do you need to take biopsies after AI endoscopy?
This article is interesting. The idea of artificial intelligence is well-known in colonoscopy, either screening or therapeutic. Much work is needed to use AI in GI endoscopy better and limit false positive results. The word fusion needs more clarification. Compared to biopsy, do you need to take biopsies after AI endoscopy? This article is interesting. The idea of artificial intelligence is well-known in colonoscopy, either screening or therapeutic. Much work is needed to use AI in GI endoscopy better and limit false positive results. The word fusion needs more clarification. Compared to biopsy, do you need to take biopsies after AI endoscopy?
